# MobileGUI-RL: Advancing Mobile GUI Agent through Reinforcement Learning in Online Environment

## Abstract

Recently, there has been a surge of vision-based GUI agents designed to automate everyday mobile and web tasks. These agents interpret raw GUI screenshots and autonomously decide where to click, scroll, or type, which bypasses handcrafted rules and app-specific APIs. However, most existing methods trained GUI agent in the offline environment using pre-collected trajectories. This approach limits scalability, causes overfitting to specific UI templates, and leads to brittle policies when faced with unseen environment. We present *MobileGUI-RL*, a scalable framework that trains GUI agent in online environment. MobileGUI-RL contains two key components. It (i) synthesizes a curriculum of learnable tasks through self-exploration and filtering, and (ii) adapts GRPO to GUI navigation with trajectory-aware advantages and composite rewards that balance task success and execution efficiency. Experiments on three online mobile-agent benchmarks show consistent gains, validating the effectiveness of our approach.

## 1 Introduction

Recent advances in large vision-language models (LVLMs) (Hurst et al., 2024; Anthropic, 2025; Bai et al., 2025) have opened up new possibilities for building vision-based GUI agents (Qin et al., 2025; Xu et al., 2025), fundamentally transforming the way intelligent agents interact with graphical user interfaces (GUIs). Unlike traditional pipeline-based GUI agents, which typically decompose the task into separate planning and GUI grounding stages (Zheng et al., 2024; Gou et al., 2025), these vision-based GUI agents leverage the powerful perception and reasoning abilities of LVLMs to directly interpret GUI screenshots and autonomously determine actions such as clicking, scrolling, and typing (Wang et al., 2024; Zhang et al., 2024a). By eliminating the need for handcrafted rules or access to underlying application APIs, vision-based GUI agents offer a flexible, scalable, and platform-agnostic solution for automating interactions across a wide range of apps and devices.

Despite these advances, training GUI agents that can operate robustly in real-world environments remains a highly challenging task. Most existing method train GUI agents in the offline environment that rely on static, pre-collected trajectory data for supervised fine-tuning (Wu et al., 2024; Qin et al., 2025; Sun et al., 2025). Another line of research explores step-wise reinforcement learning, inspired by the recent DeepSeek-R1 paradigm (Zhou et al., 2025; Lu et al., 2025b; Luo et al., 2025). However, offline learning methods rely extensively on high-quality annotations for action trajectories, which require step-by-step executions and precise evaluations of their correctness. Such detailed annotations are labor-intensive and challenging to scale (Wu et al., 2024; Qin et al., 2025). Moreover, GUI agents trained with SFT or offline reinforcement learning often overfit to specific interface patterns (Qin et al., 2025; Sun et al., 2025; Xu et al., 2025). Such overfitting leads to poor generalization when encountering task instructions deviating from familiar templates or to dynamic UI environments. In practice, real-world GUIs are highly variable: new screens frequently emerge, interface elements change or disappear unpredictably, and user interactions can substantially alter GUI states. Pre-trained policies struggle to adapt to such changes, limiting real-world usability.

To address these limitations, training GUI agents in online environments has emerged as a promising direction (Bai et al., 2024; Wang et al., 2025), enabling agents to continuously interact with their environment and update policies in real time. However, it introduces several challenges. First, online

Figure 1: Framework overall – a scalable pipeline for training GUI agents through self-exploration, task filtering, and trajectory-level reinforcement learning with a structured reward design.

learning requires real-time interaction with the environment at every training step. Each action must be executed, and its effect observed, before updating the policy. This process can be slow and computationally expensive, especially when scaling to complex apps or mobile devices where GUI rendering and response times vary. Second, defining meaningful trajectory-level reward signals is nontrivial. Many tasks have long trajectories, where the agent must execute a sequence of steps before achieving a goal. At the same time, multiple action sequences may lead to the same outcome, and near-correct trajectories can fail due to a single misstep. These challenges make reward-driven learning difficult, potentially slowing convergence and leading to suboptimal policies.

In this work, we present MobileGUI-RL, a novel framework for training GUI agents through reinforcement learning in online environments. To support this, we develop an interactive environment that supports virtual machine management and continuous online learning, enabling agents to explore and adapt to the full spectrum of mobile GUI interactions. MobileGUI-RL consists of two key components. First, we employ a synthetic task generation pipeline that combines self-exploration with filtering, producing a curriculum of learnable tasks tailored to the agent's current capabilities. Then, we adapt group relative policy optimization (GRPO) (Shao et al., 2024; Guo et al., 2025) and introduce a trajectory-aware advantage and multi-component rewards that balance task success and execution efficiency. Experiments on four mobile agent benchmarks show that MobileGUI-RL improves performance in both online and offline evaluations. We also observe a steady improvement in online performance throughout the reinforcement learning process.

## 2 RELATED WORK

**GUI Agent** The paradigm for autonomous GUI interaction has recently shifted towards agents powered by large vision language models (LVLMs) as their core reasoning engine (Gur et al., 2023; Zhang et al., 2024a; Wang et al., 2024). These agents typically interpret visual data from screenshots, sometimes augmented with structural information, to predict their actions. Earlier approaches often employed multi-stage pipelines with separate planning and grounding phases Zheng et al. (2024); Gou et al. (2025). However, the field is steadily shifting toward end-to-end models that operate directly on raw pixels, offering a more scalable and human-like framework for UI interaction (Hong et al., 2024; Xu et al., 2025; Qin et al., 2025). While recent methods have enhanced agent reasoning, navigating complex and dynamic mobile environments still demands more advanced planning and adaptability. For example, Aguvis (Xu et al., 2025) augments datasets with language model generated chain-of-thought annotations, while UI-TARS (Qin et al., 2025) incorporates both positive and negative examples to support self-reflection and error correction via direct preference optimization. Other approaches improve navigation and error recovery by giving agents explicit control over their

trajectories—such as the ability to rollback to prior states or explore alternative action paths (Zhang et al., 2025; Hu et al., 2025). To address stagnation in self-improving agents, WebEvolver (Fang et al., 2025) introduces co-evolving world models for look-ahead simulation. In contrast to prior work that relies on offline tuning or static environments, MobileGUI-RL adopts online reinforcement learning from live interaction trajectories, enabling real-time policy updates and more adaptive performance.

**RL with Agent**    Recent advancements in GUI agents have marked a significant shift from reliance on supervised fine-tuning (SFT) to the adoption of reinforcement learning (RL) to improve generalization and decision-making capabilities. This trend is largely inspired by the success of DeepSeek-R1 (Guo et al., 2025). The "R1-style" training paradigm, which uses RL to directly optimize policies based on task rewards, has been effectively applied to the agent domain. For example, GUI-R1 demonstrated that RL can achieve state-of-the-art performance on GUI tasks across multiple platforms while using only a fraction of the data required by SFT methods (Luo et al., 2025). Similarly, WebAgent-R1 employed a multi-turn RL framework for web navigation, significantly boosting base model success rates by learning directly from online interactions with binary success signals (Wei et al., 2025).

While the direct application of RL has proven effective, the unique challenges of long-horizon interaction, sparse rewards, and high data cost inherent to GUI and web agent tasks have spurred the development of more specialized algorithms. To improve credit assignment in multi-step tasks, GiGPO improves credit assignment with hierarchical advantage estimation (Feng et al., 2025), while ARPO enhances GRPO with replay buffers for better sample efficiency (Lu et al., 2025a). To reduce reliance on human-annotated data, DigiRL adopts an autonomous offline-to-online RL pipeline (Bai et al., 2024). Complementary efforts like ZeroGUI use VLMs for automatic task and reward generation (Yang et al., 2025), and InfiGUI-R1 evolves agents from reactive to deliberative through RL-based planning and recovery (Liu et al., 2025).

## 3 MOBILEGUI-RL

### 3.1 OVERVIEW

We formulate the problem of GUI task completion as a Markov Decision Process (MDP), defined by the tuple $\mathcal{M} = (\mathcal{S}, \mathcal{A}, \mathcal{P}, \mathcal{R})$ (Fang et al., 2025; Hu et al., 2025). Here, $\mathcal{S}$ represents the state space of GUI screenshots and system states, $\mathcal{A}$ encompasses the action space of user interactions (e.g., taps, swipes, text input), $\mathcal{P}$ denotes the transition determined by the mobile operating system, $\mathcal{R}$ is a reward function that evaluate task completion. Given a natural language instruction $\mathbf{q}$, our goal is to train an agent to learn a policy $\pi_\theta(\mathcal{A} \mid \mathcal{S}, \mathbf{q})$ to complete the given task accurately while maximizing the expected cumulative reward over time. This MDP formulation provides a principled framework for learning and evaluating interactive agents in complex, dynamic mobile GUI environments.

To train GUI agents using online trajectory reinforcement learning, we propose three novel modules, detailed as follows: First, we design an interactive environment that supports continuous online learning, enabling agents to explore and adapt across the full spectrum of mobile GUI interactions (Section 3.2). Second, we introduce a synthetic task generation pipeline that combines self-exploration with task filtering, yielding a dynamic curriculum tailored to the agent's evolving capabilities (Section 3.3). Third, we adapt Group Relative Policy Optimization (GRPO) to the unique challenges of GUI navigation, incorporating trajectory-aware advantage estimation and a multi-component reward structure that balances task success with execution efficiency (Section 3.4).

### 3.2 SCALABLE AND INTERACTABLE ENVIRONMENT FOR ONLINE LEARNING

To support GUI agents with online trajectory reinforcement learning, we design a training environment centered on two key capabilities: *batched virtual execution* and *real-time agent interaction*.

**Batched Virtual Execution.**    At the core of our system is a scalable, asynchronous framework that deploys multiple Android emulator (Android Developers, 2024) instances in parallel across CPU machines. This batched execution enables agents to interact with diverse GUI environments simultaneously, significantly increasing throughput and trajectory diversity. Trajectories are collected asynchronously from a pool of emulators running in parallel, while policy optimization is performed separately on GPU servers. This architecture aligns compute-intensive environment simulation with

CPU resources and model training with GPUs, optimizing overall resource utilization. Moreover, the asynchronous design improves scalability, allowing more environment instances to be added without bottlenecking training, even when hardware performance varies. As a result, the framework supports large-scale rollout and yields more diverse interaction trajectories, enhancing the robustness and generalization of the learned policies across real-world mobile GUI tasks.

**Real-Time Agent Interaction.** At each timestep $t$, the agent observes a multimodal state representation $s_t = (v_t, \mathbf{q}, \mathbf{h}_t)$ comprising three essential components (Zheng et al., 2024). The visual input $v_t$ provides the current screenshot capturing the complete GUI state. The task goal $\mathbf{q}$ specifies the natural language instruction that guides the agent's behavior. The interaction history $\mathbf{h}_t = \{(s_0, a_0), ..., (s_{t-1}, a_{t-1})\}$ maintains temporal context, enabling the agent to reason about past actions and their consequences. More details on input construction are in appendix **X**.

The agent, a vision-language model in our setting, will process this state representation and first generate an internal reasoning trace $\mathbf{c}_t$, then produce a structured action $a_t \in \mathcal{A}$. Our action space comprehensively covers mobile interactions through four categories: (1) Physical gestures include parameterized actions such as $\text{tap}(x, y)$ and $\text{swipe}(x_1, y_1, x_2, y_2)$ using normalized coordinates $\in [0, 1]^2$ for resolution independence; (2) Text input actions $\text{type}(\text{string})$ handle keyboard interactions; (3) System navigation encompasses device-level operations including $\{\text{back}, \text{home}, \text{recent}\}$; (4) Control actions include $\text{wait}(t)$ for synchronization with dynamic UI elements and $\text{terminate}(\text{status})$ for episode completion. More detailed action definitions are provided in the appendix section.

A sequence of these interactions $\tau = (s_0, a_0, s_1, a_1, ..., s_T)$ forms a *trajectory*, which is evaluated upon termination. Rather than relying on hand-crafted reward functions that poorly generalize across tasks, we employ a powerful vision-language model oracle $\mathcal{O}$ (e.g., Qwen 2.5 VL 72B) to serve as a unified evaluator. Given the final $k$ screenshots of a trajectory and the initial instruction $\mathbf{q}$, the oracle analyzes whether the task has been completed as intended: $r = \mathcal{O}(\{s_{T-k+1}, ..., s_T\}, \mathbf{q})$ (Bai et al., 2024). This evaluation produces a binary success signal that abstracts away low-level UI details, enabling scalable supervision across diverse GUI tasks without task-specific engineering.

### 3.3 SYNTHETIC TASK GENERATION AND FILTERING

A critical challenge in training GUI agent in online environment is obtaining a diverse yet learnable curriculum of tasks. Real-world task distributions are heavily skewed toward common interactions, limiting the agent's ability to handle edge cases. Moreover, manually curating tasks is labor-intensive and fails to scale with the complexity of modern mobile ecosystems. We address this through a two-stage pipeline that automatically generates and filters synthetic tasks.

#### 3.3.1 SELF-EXPLORATION FOR DIVERSE TASK DISCOVERY

Our self-exploration mechanism leverages the natural structure of mobile interfaces to discover meaningful tasks. The process begins with an exploration agent $\pi_{\text{explore}}$ performing random walks through the GUI environment. These walks are not purely random but incorporate basic heuristics such as preferring unexplored UI elements and avoiding repetitive loops. Each exploration trajectory $\tau_{\text{explore}} = \{(s_0, a_0), ..., (s_n, a_n)\}$ captures a sequence of state transitions that potentially represent a coherent task. Inspired by Sun et al. (2025), we then employ GPT-4o to reverse-engineer task descriptions from these trajectories. Given a trajectory, the model generates a natural language instruction $\mathbf{q}$ that would motivate the observed sequence of actions. This reverse process – figuring out the goal from the actions – produces a variety of tasks that match what the app is designed to do. The generated tasks span a wide spectrum, from simple interactions ("Open the settings menu") to complex multi-step procedures ("Set a recurring alarm for weekdays at 7 AM").

#### 3.3.2 TASK FILTERING VIA TEXT-BASED WORLD MODEL

While self-exploration can generate a wide range of task instructions, many of them suffer from two key issues: they are either too ambiguous, often due to limitations in the reverse-engineered LLM's summarization ability, or too complex to be solved given the current GUI state and context. Attempting to execute these infeasible tasks leads to wasted computational effort and the generation of low-quality trajectories that may destabilize learning. To address this, we propose a lightweight filtering mechanism based on a text-based world model, which pre-screens candidate tasks before

rollout. This approach effectively avoids unnecessary environment interactions while ensuring that the selected tasks are within the agent's current capability.

Our filter first employs a LLM as a simulator $\mathcal{W}$ that can generate a textual representation $\tilde{s}$ of the GUI state. Given a task $\mathbf{q}$ and current state description $\tilde{s}$, the world model predicts the next state $\tilde{s}' = \mathcal{W}(\tilde{s}, a, \mathbf{q})$ resulting from action $a$. The filtering proceeds as follows: The world model first initializes with a textual description of the home screen, structured as a list of UI elements with their properties: $\tilde{s}_0 = \{e_1 : (\text{type, content, bounds}), ..., e_n : (\text{type, content, bounds})\}$. Our base agent $\pi_{\text{base}}$ receives both the task and this description and outputs an action. The world model simulates the action's effect, generating a new state description that reflects the expected GUI changes. This process continues until the base agent signals task completion or exceeds a step limit $T_{\max}$.

A task is admitted to the training set only if the simulation reaches a success state within the step limit: $\mathcal{F}(\mathbf{q}) = \mathbb{1}[\exists t \leq T_{\max} : \pi_{\text{proxy}}(a_t|\tilde{s}_t, \mathbf{q}) = \text{terminate(success)}]$. This filtering process serves not only to remove logically inconsistent or overly complex tasks, but also plays a key role in decoupling perception from reasoning. Since the world model operates entirely on structured textual representations of the GUI, it removes the need for low-level visual grounding. As a result, the evaluation focuses solely on whether the agent's reasoning and planning abilities are sufficient to solve the task, assuming perfect perception. This abstraction allows us to efficiently assess task feasibility and construct a curriculum without being confused by perception errors.

## 3.4 ONLINE LEARNING WITH MOBGRPO

Training GUI agent in online environments presents several unique challenges. Rewards are typically sparse, trajectories can be long with variable step counts, and task outcomes often depend on delayed success signals. These challenges make credit assignment difficult and destabilize training under standard policy gradient methods such as PPO (Schulman et al., 2017). To address these issues, we extend GRPO (Shao et al., 2024) with a trajectory-aware formulation and a carefully designed reward structure for mobile GUI agent training, forming our proposed *MobGRPO* algorithm.

### 3.4.1 TRAJECTORY-AWARE POLICY OPTIMIZATION

Our MobGRPO objective builds upon GRPO to handle variable-length trajectories and fine-grained action steps. For a batch of $G$ trajectories $\{\tau_i\}_{i=1}^{G}$ generated for task $\mathbf{q}$, we define the loss as:

$$\mathcal{L}_{\text{MobGRPO}} = -\frac{1}{\sum_{t=1}^{G} |\mathbf{o}_{i,s}|} \sum_{i=1}^{G} \sum_{s=1}^{S_i} \sum_{t=1}^{|\mathbf{o}_{i,s}|} \left\{ \min \left[ r_t(\theta) \hat{A}_{i,s,t}, \, \text{clip}\left(r_t(\theta), 1-\epsilon, 1+\epsilon\right) \hat{A}_{i,s,t} \right] \right\} \quad (1)$$

where $r_t(\theta) = \frac{\pi_\theta(o_{s,t}|s_{<s}, a_{<s}, o_{s,<t})}{\pi_{\theta_{\text{old}}}(o_{s,t}|s_{<s}, a_{<s}, o_{s,<t})}$ is the token-level probability ratio in the action sequence, and $\hat{A}_{i,s,t}$ shares a trajectory-level advantage signal. Instead of computing per-step rewards and advantages, we evaluate the entire trajectory $\tau$ after completion to obtain a single scalar reward $R(\tau, \mathbf{q})$ that reflects its overall quality. This trajectory-level reward is then used to compute a normalized advantage: $\hat{A}_\tau = \frac{R(\tau, \mathbf{q}) - \bar{R}_{\mathbf{q}}}{\sigma_{R_{\mathbf{q}}} + \epsilon}$, where $\bar{R}_{\mathbf{q}}$ and $\sigma_{R_{\mathbf{q}}}$ denote the mean and standard deviation of trajectory rewards for task $\mathbf{q}$. This advantage is uniformly assigned to all steps within the trajectory, providing a consistent learning signal regardless of the trajectory length or where the success occurs.

By aggregating the reward at the trajectory level and distributing the advantage across all steps, our approach avoids noisy or misleading per-step supervision and addresses the credit assignment problem in long-horizon GUI tasks.

### 3.4.2 MULTI-COMPONENT REWARD DESIGN

Reward design plays a central role in learning effective policies for GUI navigation, where tasks are long-horizon, rewards are sparse, and outcomes are often binary. Standard reward functions, e.g., assigning $r = 1$ for success and $r = 0$ otherwise, are inadequate in this context. They fail to differentiate between successful trajectories of varying quality and offer no learning signal when all rollouts in a batch succeed or fail uniformly. To address these limitations, we propose a multi-component reward function that captures trajectory-level quality, discourages premature termination, and ensures continuous learning signals for policy optimization.

Table 1: Agent action space for GUI interaction.

| Action Type | Description |
|---|---|
| click | Tap at a specified (x, y) coordinate. |
| swipe | Swipe from a start coordinate to an end coordinate. |
| type | Input specified text into the active UI element. |
| system_button | Press a system-level button (e.g., Back, Home). |
| wait | Pause execution for a specified number of seconds. |
| terminate | End the task, declaring final success or failure. |
| answer | Provide a textual response for question-answering tasks. |

**Differentiating Successful Trajectories.** Although many trajectories may successfully complete a task, they can differ significantly in terms of efficiency. In GUI settings, shorter trajectories are generally preferred as they reduce user friction and lower the risk of compounding errors. To reflect this, we introduce an exponentially decaying efficiency factor that rewards faster completions more,

$$f_{\text{efficiency}}(|\tau|) = \text{clip}(e^{-\lambda|\tau|}, \alpha_{\min}, \alpha_{\max}). \tag{2}$$

This design not only encourages efficient behavior but also addresses a key issue in GRPO-like methods: when all trajectories succeed and receive identical rewards (e.g., $r = 1$), the normalized advantage becomes zero, halting policy updates. Our reward structure introduces relative differences even among successful rollouts, preserving gradient signals for continued learning.

**Penalizing Premature Termination.** Agents trained on sparse-reward environments often learn to "give up" early when facing difficult or ambiguous tasks, terminating episodes prematurely before fully attempting or exploring the instruction. To discourage this behavior, we introduce a penalty for early exits when the task is not yet completed:

$$g(|\tau|) = 1 - \frac{|\tau|}{T_{\max}}, \quad \text{penalty} = \beta_{\max} \cdot g(|\tau|) \tag{3}$$

This linear decay penalizes early termination more heavily than later exits, encouraging the agent to engage more thoughtfully with the task before deciding to stop.

**Handling Degenerate Batches.** Another practical issue arises when all trajectories in a batch fail, yielding zero rewards. In such cases, the computed advantages are uniformly zero, resulting in no policy update. We adopt the same mitigation as proposed in DAPO (Yu et al., 2025) by filtering out these degenerate batches during training to maintain meaningful optimization dynamics.

**Final Reward Formulation.** Combining these components, our composite reward is defined as:

$$R(\tau, \mathbf{q}) = \begin{cases} r_{\text{base}} \cdot f_{\text{efficiency}}(|\tau|) & \text{if success} \\ -\beta_{\max} \cdot g(|\tau|) & \text{if fail} \end{cases} \tag{4}$$

This formulation delivers a dense, interpretable, and differentiable learning signal that encourages success, promotes efficiency, penalizes shortcuts, and maintains update dynamics across varying batch conditions. The modular design also allows fine-tuning through hyperparameters to suit deployment-specific requirements.

## 4 EXPERIMENTS

### 4.1 EXPERIMENTS SETTING

**Parameter Settings**  Our training environment is built on a scalable pool of Android Virtual Devices (AVDs), with the exact number determined by the batch size. Each AVD runs on an emulated device with a 1080×2400 resolution. We train our agent for one epoch on a dataset of 436 curated GUI navigation tasks. For each task, we collect eight rollouts using 7B models and four rollouts using 32B models, with a maximum episode length of 25 steps. For a comprehensive list of all environment and training hyperparameters, please refer to the Appendix in Section A.

Table 2: Performance on GUI Agent Benchmarks. We report results across three online mobile GUI agent benchmarks, evaluating each method by Success Rate (SR).

| Models | AW (SR) | AITW-Gen (SR) | AITW-Web (SR) |
|---|---|---|---|
| *Closed-source Models* | | | |
| GPT-4o (Hurst et al., 2024) | 34.5 | - | - |
| Claude Computer Use (Anthropic, 2024) | 27.9 | - | - |
| *Open-source 7B Models* | | | |
| OS-Genesis-7B (Sun et al., 2024) | - | 0.7 | 0.0 |
| OS-Atlas-7B (Wu et al., 2024) | - | 15.7 | 17.3 |
| Aguvis-7B (Huang et al., 2024) | - | 23.0 | 4.7 |
| Qwen2.5-VL-7B (Bai et al., 2025) | 22.0 | 49.0 | 20.0 |
| UI-TARS-7B (Qin et al., 2025) | 33.0 | 48.0 | 16.7 |
| **MobileGUI 7B (Ours)** | 30.0 | **65.3** | 22.7 |
| *Open-source 32B/72B Models* | | | |
| Qwen2.5-VL-32B (Bai et al., 2025) | 31.5 | 42.7 | 24.7 |
| Qwen2.5-VL-72B (Bai et al., 2025) | 35.0 | 51.3 | **31.3** |
| Aguvis-72B (Huang et al., 2024) | 26.1 | - | - |
| **MobileGUI 32B (Ours)** | **44.8** | 58.0 | 30.7 |

**Agent Construction**  Our agent, **MobileGUI**, is built upon the `Qwen2.5-VL-7B-Instruct` and `Qwen2.5-VL-32B-Instruct` large multi-modal model. It processes both visual information from screenshots and textual task descriptions. To interact with the environment, the agent uses a structured tool-use interface, where it generates actions by calling a predefined `mobile_use` function. The agent is prompted to first externalize its reasoning within `<thinking>` tags and then generate a valid action call. The prompt provides the function signature, outlining the available action types and their required parameters. A detailed description of the prompt is available in Appendix B. The agent's action space, as defined in the function signature, is summarized in Table 1.

**Baselines and Benchmarks**  We evaluate on three online GUI agent benchmarks that require agents to complete a variety of tasks within interactive environments. The evaluation spans three benchmark settings: **AndroidWorld (AW)** (Rawles et al., 2024), **Android-in-the-Wild General Tasks (AITW-Gen)**, and **Android-in-the-Wild WebShop (AITW-Web)** (Zhang et al., 2024b). Performance is measured using several metrics, including **Success Rate (SR)**, which reflects the proportion of tasks successfully completed. Our results are compared against a range of state-of-the-art closed-source and open-source models, including GPT-4o, Claude Computer Use, and other notable open-source VLMs like Qwen2.5-VL and OS-Atlas. The detailed performance comparison is presented in the results section.

## 4.2  MAIN RESULTS

We evaluate our MobileGUI-RL framework by applying it to two powerful base models, Qwen2.5-VL-7B and Qwen2.5-VL-32B, creating our MobileGUI-7B and MobileGUI-32B agents. The performance of our models against state-of-the-art closed-source and open-source baselines is detailed in Table 2. As presented in Table 2, our MobileGUI-RL framework delivers substantial performance enhancements to the base models across all three benchmarks. The results underscore the efficacy of online reinforcement learning for improving GUI navigation capabilities in large multi-modal models.

Our smaller model, MobileGUI-7B, demonstrates significant gains over its base model, Qwen2.5-VL-7B. The Success Rate (SR) on AndroidWorld (AW) improves from 22.0% to 30.0%, and most notably, we see a remarkable jump on AITW-Gen tasks from 49.0% to 65.3%. This represents a 16.3 point improvement. While UI-TARS-7B achieves a higher SR on AW (33.0% vs 30.0%), it's important to note that UI-TARS required training on over 50 billion tokens for continued pre-training, whereas our approach achieves competitive performance using a substantially smaller dataset of only 436 curated tasks. This demonstrates the remarkable data efficiency of our online reinforcement learning framework. Moreover, our model's dominant performance on AITW-Gen (65.3% vs 48.0%) highlights its superior ability to generalize to diverse, real-world scenarios.

The most compelling results are observed with our larger model. **MobileGUI-32B** boosts the performance of its base model, Qwen2.5-VL-32B, by 13.3 points. On the challenging AndroidWorld benchmark, our model achieves an SR of **44.8%**, decisively outperforming all other baseline models, including the leading closed-source model GPT-4o (34.5%) and the much larger Qwen2.5-VL-72B (35.0%). This demonstrates that our RL fine-tuning method is not only effective but also highly efficient, enabling a 32B model to surpass a 72B model from the same family. In addition, MobileGUI-32B achieves strong performance on AITW-Gen (58.0%) and AITW-Web (30.7%), showing consistent and robust gains across diverse task distributions. In summary, our MobileGUI-RL framework makes consistent and significant performance gains. The improvements are particularly pronounced in the larger model, suggesting that our online learning approach effectively refines the model's existing capabilities to master complex GUI interaction tasks.

## 4.3 ABLATION STUDY

We conduct a series of ablation studies to systematically evaluate the contribution of each key component within our MobileGUI-RL framework. Specifically, we investigate the impact of: (1) our text-based world model for task filtering, (2) the implicit curriculum learning derived from it, and (3) our multi-component decaying reward function. We benchmark all variants on the Android World (AW) dataset, and the results are summarized in Table 3.

Table 3: Ablation study of our key components on the Android World (AW) benchmark. We report the task success rate (%). Our full MobileGUI-RL model significantly outperforms variants where a key component is removed, demonstrating the effectiveness of each design choice.

| Configuration | 7B Model (AW %) | 32B Model (AW %) |
| --- | --- | --- |
| MobileGUI-RL (Full Model) | 30.0 | 44.8 |
| w/o Task Filtering | 28.5 | 41.0 |
| w/o Curriculum Learning | 25.0 | 34.0 |
| w/o Decaying Reward | 23.5 | 35.5 |

**The Effect of Task Filtering.** To validate the effectiveness of our task filtering mechanism, we compare our full model against a variant trained on the complete, unfiltered set of synthetically generated tasks. Our self-exploration phase initially produced 1251 candidate tasks. Our text-based world model filter pruned this set to 436 tasks deemed solvable and unambiguous. As shown in Table 3, removing this filter leads to a substantial performance degradation of 1.5 and 3.8 percentage points for the 7B and 32B models, respectively. This highlights the critical importance of filtering. Training on the unfiltered set exposes the agent to a high volume of low-quality or unsolvable tasks, which introduces significant noise into the learning process. This forces the agent to waste computational resources on unproductive trajectories, ultimately destabilizing policy optimization and resulting in a less capable final agent.

**The Effect of Curriculum Learning.** Our task generation pipeline implicitly creates a curriculum by estimating task complexity via the number of steps required for completion in the text-based world model. To ablate its effect, we trained a model on the same filtered task set but sampled tasks uniformly at random, removing the complexity-based ordering. The results, summarized in Table 3, demonstrate a substantial performance drop when the curriculum is removed. The 7B model's success rate falls by 5 points (from 30.0% to 25.0%), and the 32B model's performance drops by a significant 10.8 points (from 44.8% to 34.0%). This highlights the curriculum's critical role in achieving high final performance.

The training dynamics in Figure 4 reveal why curriculum learning is more effective. The curriculum approach (red lines) provides dense, positive reward signals early in training when the model can successfully complete simpler tasks, enabling stable learning of fundamental interaction skills. As training progresses and harder tasks are introduced, the reward naturally decreases and impossible task ratios rise, but by this point the model has already been properly trained on core capabilities. This sequencing maximizes data efficiency by ensuring the model learns from easy tasks before tackling more difficult ones.

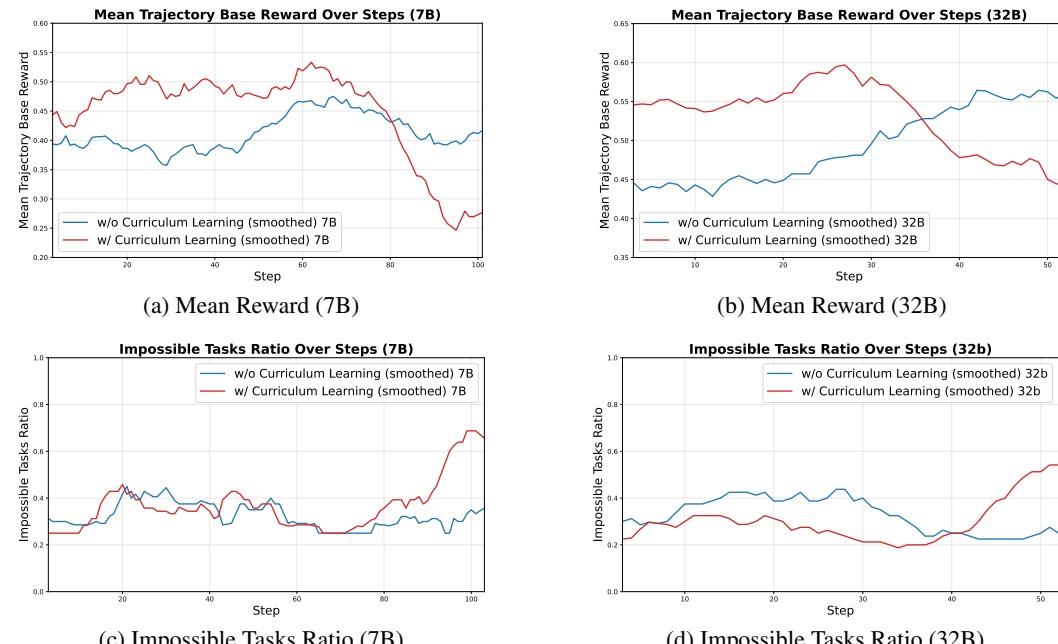

(a) Mean Reward (7B)          (b) Mean Reward (32B)

(c) Impossible Tasks Ratio (7B)          (d) Impossible Tasks Ratio (32B)

Figure 2: Training dynamics with and without curriculum learning for 7B and 32B models. The top row shows the mean trajectory base reward, and the bottom row shows the ratio of impossible tasks encountered. With curriculum learning (red), the reward first rises on easy tasks and then falls as the curriculum introduces harder tasks, which is corroborated by the rising impossible task ratio. This structured approach leads to better final performance than training without a curriculum (blue).

In contrast, uniform sampling (blue lines) exposes the model to a mixture of easy and hard tasks from the start. While this produces more stable-looking curves, it wastes valuable learning opportunities by presenting unsolvable tasks before the model has mastered basics. The superior final performance of our curriculum approach (Table 3) demonstrates that strategic data ordering, not curve stability, drives effective learning.

**The Effect of Decaying Reward.**   Finally, we evaluate the effectiveness of our multi-component reward design, focusing on the exponential decay factor that encourages efficiency. We compare our full model against a variant using a simple binary reward ($r = 1$ for success, $r = 0$ otherwise). Removing the decaying component leads to significant performance drops of 6.5 and 9.3 points for the 7B and 32B models, respectively, highlighting the limitations of sparse, binary rewards in complex GUI navigation tasks. The decaying reward plays two key roles. First, it introduces reward variance among successful trajectories, motivating the agent to seek not just correct, but efficient solutions. Second, it mitigates a common failure mode in GRPO-style algorithms: when all trajectories in a batch succeed with identical rewards, the normalized advantage becomes zero, halting learning. By preserving gradient signals and rewarding efficiency, our reward formulation provides dense, informative feedback that is critical for effective policy optimization.

## 5 CONCLUSION

We present MobileGUI-RL, a reinforcement learning framework for training GUI agents in online environments. By creating an interactive setup, our framework enables agents to adapt to dynamic and unpredictable mobile UIs. To address data generation, we introduce a synthetic task pipeline that combines self-exploration with a text-based world model for curriculum filtering. Our MobGRPO algorithm further incorporates trajectory-aware advantages and multi-component rewards, optimizing both task success and efficiency. Experiments show that fine-tuned MobileGUI agents achieve substantial gains on challenging GUI benchmarks. In particular, MobileGUI-32B outperforms its base model and leading closed-source competitors, demonstrating that online reinforcement learning with trajectory-level feedback is a powerful paradigm for building robust and capable GUI agents.

## ETHICS STATEMENT

This research utilizes the Qwen foundation model Bai et al. (2025), operating within the scope of its academic licensing agreement. Our implementation strictly adheres to the academic-use provisions specified in the license, with all applications limited to scholarly research purposes. The study draws upon two datasets: AndroidWorld Rawles et al. (2024) and Android-in-the-Wild (AITW) Bai et al. (2024), each employed in accordance with their respective usage guidelines and data governance frameworks. We have conducted thorough reviews to ensure compliance with data protection protocols. Furthermore, our data processing protocols have verified that the content is appropriate and free from inappropriate material, maintaining high standards of research ethics and data integrity.

## REPRODUCIBILITY STATEMENT

We have taken several steps to facilitate reproducibility. The full algorithmic specification of MOBILEGUI-RL and its training objective are described in Section 3.4, with the action space summarized in Table 1 and ablations supporting design choices in Table 3. Exact environment and training hyperparameters (AVD configuration, batch sizing, optimization settings, rollout limits) are enumerated in Appendix A; the prompting and tool-use interface needed to reproduce agent behavior are provided verbatim in Appendix B. Dataset usage and evaluation protocols (including curation steps for AITW-Gen/Web and adherence to AndroidWorld procedures) are documented in Appendix C.

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

# A DETAILED TRAINING CONFIGURATION

## A.1 ENVIRONMENT SETUP

### A.1.1 ANDROID EMULATOR CONFIGURATION

- **Base AVD Name**: AndroidWorldAvd
- **Emulator Instances**: Dynamically scaled based on batch size
- **Screen Resolution**: 1080×2400 pixels
- **Memory Allocation**: 3072 MB per emulator
- **CPU Cores**: 2 cores per emulator
- **GPU Acceleration**: Auto mode

## A.2 MODEL AND TRAINING HYPERPARAMETERS

### A.2.1 MODEL CONFIGURATION

- **Base Model**: Qwen2.5-VL-7B-Instruct and Qwen2.5-VL-32B-Instruct
- **Attention Implementation**: Flash Attention 2
- **Gradient Checkpointing**: Enabled
- **Mixed Precision**: BFloat16 for parameters, FP32 for reduction

### A.2.2 GRPO TRAINING PARAMETERS

- **Global Batch Size**: 128
- **Micro Batch Size (Update)**: 4 per device
- **Micro Batch Size (Experience)**: 16 per device
- **Learning Rate**: $1 \times 10^{-6}$
- **Adam Betas**: (0.9, 0.999)
- **Weight Decay**: 0.01
- **Gradient Clipping**: 1.0
- **PPO Clip Ratio**: 0.2
- **Entropy Coefficient**: $1 \times 10^{-3}$
- **KL Penalty Coefficient**: $1 \times 10^{-2}$ (for GRPO)
- **PPO Epochs**: 1
- **Advantage Estimator**: GRPO with trajectory-based normalization

### A.2.3 ROLLOUT CONFIGURATION

- **Temperature**: 1.0
- **Top-p**: 1.0
- **Max Response Length**: 2048 tokens
- **Number of Rollouts per Prompt**: 8
- **Maximum Steps per Episode**: 15
- **Tensor Parallel Size**: 2
- **GPU Memory Utilization**: 0.5

## B PROMPT CONSTRUCTION

The agent operates through a structured tool-use interface. The system prompt provides the agent with a function signature for mobile device interaction:

```
1  {
2    "type": "function",
3    "function": {
4      "name": "mobile_use",
5      "description": "Use a touchscreen to interact...",
6      "parameters": {
7        "properties": {
8          "action": {
9            "enum": ["click", "swipe", "type",
10                     "system_button", "wait",
11                     "terminate", "answer"]
12          },
13          "coordinate": {"type": "array"},
14          "coordinate2": {"type": "array"},
15          "text": {"type": "string"},
16          "time": {"type": "number"},
17          "button": {"enum": ["Back", "Home",
18                              "Menu", "Enter"]},
19          "status": {"enum": ["success", "failure"]}
20        }
21      }
22    }
23  }
```

The agent is instructed to provide reasoning within `<thinking>` tags before each action and summarize actions within `<conclusion>` tags. Task progress is tracked by maintaining a history of previous actions and their outcomes.

The evaluator provides binary success/failure judgments along with detailed reasoning about whether all task requirements have been satisfied.

## C EVALUATION DETAILS

This section provides additional details on the evaluation procedures for each benchmark used in our experiments.

**AndroidWorld**   For the AndroidWorld benchmark, we utilized the official evaluation code and procedures released by the original authors (Rawles et al., 2024). This ensures that our results are directly comparable to previously reported scores on this benchmark.

**Android-in-the-Wild (AITW)**   For the AITW-Gen and AITW-Web benchmarks, we adapted the evaluation scripts originally provided in the DigiRL study (Bai et al., 2024). We made several modifications to curate the datasets for our specific testing environment.

- **AITW-Gen:** We manually reviewed the tasks and removed those that could not be reliably executed on our emulated Android environment. These tasks primarily involved actions such as installing specific third-party applications, which were not feasible in our sandboxed virtual devices. After this filtering process, the final AITW-Gen dataset used for our evaluation consisted of 300 unique tasks.
- **AITW-Web:** During our review of the WebShop tasks, we identified a significant number of duplicate entries. To create a more robust and less redundant benchmark, we performed a deduplication process, merging these similar tasks. This resulted in a final, curated AITW-Web benchmark of 150 unique tasks.

These curation steps were taken to ensure a fair and consistent evaluation of the agent's capabilities on tasks that are executable within our standardized environment.

# D CASES

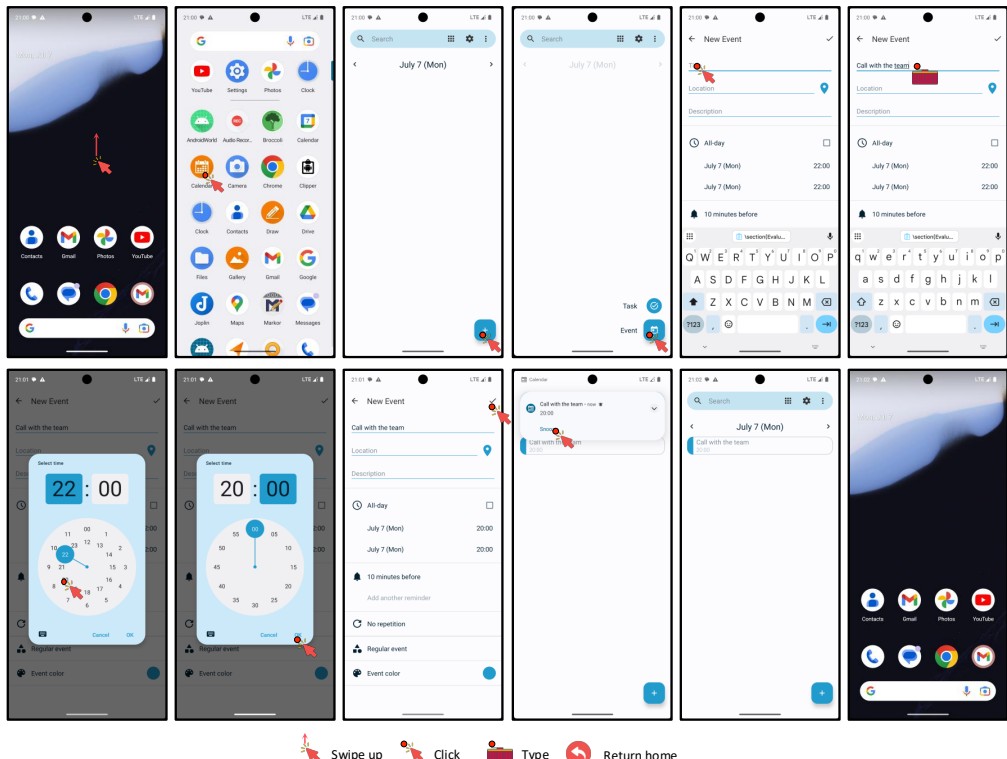

Swipe up    Click    Type    Return home

# E THE USE OF LARGE LANGUAGE MODELS (LLMS)

Large language models (LLMs), including Claude and ChatGPT, were used in a limited capacity for sentence-level editing, such as grammar and syntax corrections, clarity improvements, stylistic consistency, and minor rephrasing. They were not involved in research ideation, experimental design, data analysis, or the generation of technical content. All research concepts, methods, results, and conclusions are the original work of the authors. The LLMs served solely as language editing tools, and all revisions were reviewed and validated by the authors.

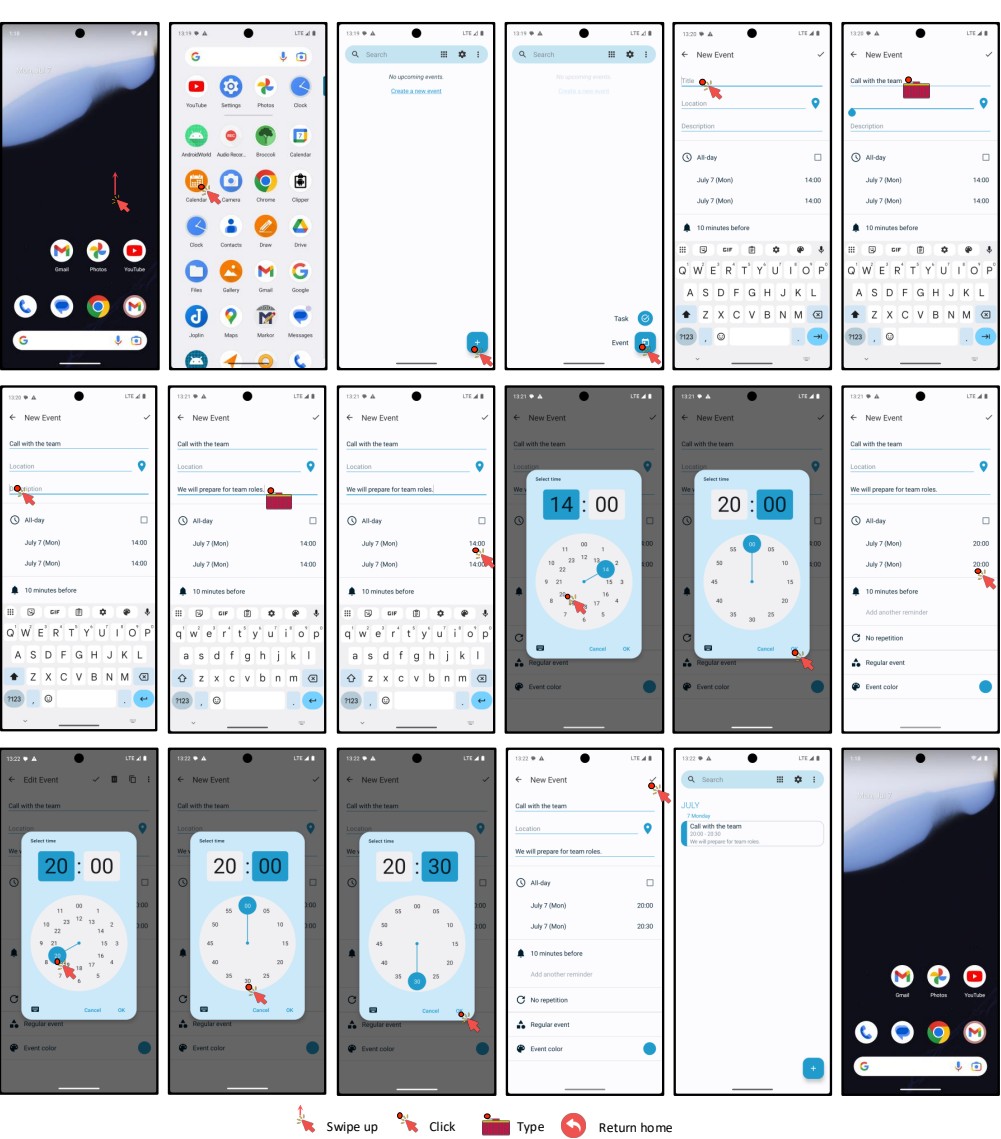

Figure 4: Case Studies. The case illustrates the task: "Create a calendar event for tomorrow at 20h with the title 'Call with the Team' and the description 'We will prepare for team roles.'. The event should last for 30 mins." The left shows the execution before reinforcement learning, while the right shows the result after RL (ours). The pre-RL agent misses two critical steps: (1) omitting the meeting description, and (2) failing to set the event's end time.

