# OpenReview forum: "MobileGUI-RL: Advancing Mobile GUI Agent through Reinforcement Learning in Online Environment"
_ICLR.cc/2026/Conference — Submitted to ICLR 2026_

### Official Review · Reviewer_ojrd · 2025-10-22

**Soundness:** 3
**Presentation:** 3
**Contribution:** 3
**Rating:** 4
**Confidence:** 5

**Summary:**

The paper *MobileGUI-RL: Advancing Mobile GUI Agent through Reinforcement Learning in Online Environment* introduces **MobileGUI-RL**, a scalable framework for training mobile GUI agents through online reinforcement learning (RL) rather than traditional offline imitation learning. The approach integrates a **powerful VLM-based unified evaluator** (e.g., Qwen2.5-VL-72B) to automatically assess success via binary visual feedback, eliminating the need for hand-crafted rewards. It also builds a diverse curriculum of mobile interaction tasks using **self-exploration** and **text-based world model filtering**, enabling the agent to learn across dynamic, realistic GUI environments.

To optimize policy learning, the authors propose **MobGRPO**, a trajectory-aware variant of GRPO that distributes a trajectory-level advantage across steps and employs a **multi-component reward** combining success, efficiency, and termination penalties. Experiments on AndroidWorld and Android-in-the-Wild benchmarks show consistent performance gains, achieving state-of-the-art results while requiring significantly fewer training tasks. The work demonstrates that online RL with automated curriculum generation and trajectory-level optimization can substantially improve generalization and robustness of mobile GUI agents.

**Strengths:**

1. The paper presents a novel online reinforcement learning framework for mobile GUI agents that departs from conventional offline imitation learning paradigms. Its integration of a *vision-language model (VLM) as a unified evaluator* for automatic reward generation, coupled with *self-exploration–based task synthesis* and *text-based world model filtering*.

2. The proposed *MobGRPO* algorithm introduces a trajectory-aware modification to GRPO with a multi-component reward structure that balances task success, efficiency, and stability. The design addresses key challenges in long-horizon, sparse-reward environments and is validated through comprehensive experiments and ablations showing consistent and interpretable improvements over strong baselines.

3. By demonstrating that online RL can outperform much larger models trained with offline data (e.g., surpassing GPT-4o and 72B VLMs using a 32B model), the work provides a strong case for efficient and scalable training of interactive agents.

**Weaknesses:**

1. Training Data Source:
In Section 4.1 Experiments Setting, the paper mentions 436 curated GUI navigation tasks. What is the construction method and coverage scope of these tasks?

2. Unified Evaluator:
Please describe the specific metrics of the evaluator, such as its correlation with SR (Success Rate) or its human verification accuracy. These details are crucial for assessing the actual contribution of the evaluator.

3. Online Sampling Efficiency:
Since online sampling for mobile agents is a computationally intensive engineering challenge, please explain the sampling efficiency and implementation details of the online learning system used in this work.

4. Reward Calculation:
The reward formulation adjusts for both short successful and failed trajectories. However, has the paper considered that excessive encouragement of shorter successful paths might reduce the policy’s exploration capability and thus harm performance on out-of-distribution (OOD) tasks?

**Questions:**

The questions are already included within the weaknesses section.

---

### Official Review · Reviewer_QBaE · 2025-10-23

**Soundness:** 2
**Presentation:** 2
**Contribution:** 3
**Rating:** 4
**Confidence:** 4

**Summary:**

This paper proposes MobileGUI-RL, a new paradigm for training GUI agents in online environments. MobileGUI-RL consists of three key components:

1. **Self-Exploration**: Utilizes GPT-4o to generate 1251 tasks through autonomous exploration.
2. **Task Filtering**: Employs a text-based world model to filter these tasks, resulting in 436 high-quality tasks.
3. **Decay Reward**: Uses Qwen2.5-VL-72B as a verifier to assign binary rewards (1 or 0). To further refine the learning signal, the binary reward is transformed into a length-aware reward that encourages shorter successful trajectories and penalizes shorter unsuccessful ones.

**Strengths:**

1. The paper presents a novel training framework for agents in online mobile environments.
2. The data processing pipeline is well-designed.
3. The 7B and 32B models achieve state-of-the-art performance on several online benchmarks.

**Weaknesses:**

1. **Writing and Organization:** The writing needs further polishing. For example:
   - "Qwen 2.5 VL 72B", "MobileGUI 7B" and "MobileGUI 32B" should be correctly formatted as "Qwen2.5-VL-72B." "MobileGUI-7B" and "MobileGUI-32B" respectively.
   - Sentences like "More details on input construction are in appendix X" should specify the actual appendix location.
   - Statements such as "More detailed action definitions are provided in the appendix section" should clearly indicate the appendix section or number.
   - I recommend refining Section 3 (Method) for better readability. For instance, provide clearer and more concise definitions for "Curriculum Learning." The section "MULTI-COMPONENT REWARD DESIGN" might be presented before "TRAJECTORY-AWARE POLICY OPTIMIZATION" for better logical flow. Consider improving ambiguous terminologies "Differentiating Successful Trajectories," "Penalizing Premature Termination," and "Handling Degenerate Batches". They read like AI-generated translations.
   - The sentence "Our agent, MobileGUI, is built upon the Qwen2.5-VL-7B-Instruct and Qwen2.5-VL-32B-Instruct large multi-modal model." is confusing; the word order and phrasing should be improved for clarity.

2. **Training Details:** More comprehensive training details are needed, such as:
   - Additional information about the online tasks used during training, including task instructions, more examples, and an analysis of training task distribution. The lack of detailed training information limits reproducibility.
   - More details in the appendix on **Curriculum Learning**, **Task-Filter**, and **Self-Exploration** procedures.

3. **Experimental Support:** The study would be strengthened by additional experiments, for example:
   - Ablation studies on data size.
   - Model ablations for components such as the verifier and task generator.

4. **Broader Benchmarks:** It is recommended to evaluate on more static benchmarks, such as AndroidControl and GUI Odyssey, if possible.

5. **Related Works:** Please include a broader discussion of related work, especially regarding task generation and length-aware reward computation.

**Questions:**

1. Could the authors clarify the differences between ARPO and MobileGUI-RL?

2. **Please address the concerns mentioned above.**

---

### Official Review · Reviewer_HP2q · 2025-10-24

**Soundness:** 3
**Presentation:** 2
**Contribution:** 2
**Rating:** 2
**Confidence:** 4

**Summary:**

This paper introduces MobileGUI-RL, a reinforcement learning framework for training GUI agents directly in online, interactive environments. MobileGUI-RL enables continuous real-time interaction, self-exploration, and policy updating, allowing agents to generalize to dynamic mobile applications. The framework combines a synthetic task generation pipeline, which uses self-exploration and a text-based world model for task filtering, with an adapted GRPO algorithm, which integrates trajectory-aware advantages and multi-component rewards to improve both task success and efficiency. Experimental results on three mobile-agent benchmarks (AndroidWorld, AITW-Gen, and AITW-Web) show that MobileGUI-RL significantly improves performance over baselines.

**Strengths:**

- This paper introduces a new approach to training GUI agents in online environments using reinforcement learning, which addresses the shortcomings of previous offline methods.
- This paper designs a scalable training infrastructure with batched virtual execution on multiple Android emulators, enabling high-throughput, asynchronous data collection. This design improves both sample efficiency and policy robustness, which is an important technical strength for large-scale deployment.
- Extensive experiments across three online benchmarks (AndroidWorld, AITW-Gen, and AITW-Web) demonstrate consistent and substantial performance gains.

**Weaknesses:**

- The study contains several details not fully disclosed. Please refer to the question section for specifics.

- The reproducibility of the results is relatively low and requires further enhancement of the experimental repeatability.

- There is a lack of more ablation studies. Please refer to the question section for details.

**Questions:**

1. What is the accuracy when using Qwen2.5 VL 72B as the evaluator?
2. If the state includes screenshots with a resolution of 1080×2400, how can the context length problem be avoided in multi-turn interactions?
3. What is $\pi_{explore}$? How does it perform random walk? How are unexpired UI elements counted, and how are element-type UI elements handled?
4. What are $\pi_{base}$ and $\pi_{world}$? How does $\pi_{world}$ ensure no hallucinations over time?
5. The task filtering and the curriculum design heavily depend on $\pi_{world}$, which relies on the performance of $\pi_{base}$ and the ability of $\pi_{world}$ to accurately predict the next state. Can you elaborate on this point?
6. RL training uses two models, 7B and 32B. Are these models trained simultaneously? The author mentions that the rollout is a combination of samples from both models. How does RL training work in this case?
7. The author designs many reward components, and the effectiveness of each component requires further experimental analysis.
9. According to the author's ablation results, applying vanilla GRPO directly to the Qwen-2.5-VL-7B model (without the author's three methods) only yields about a 1% improvement in accuracy, which seems to be below expectations. Could you provide a detailed explanation and analysis of the reasons?

---

### Official Review · Reviewer_h578 · 2025-10-30

**Soundness:** 2
**Presentation:** 2
**Contribution:** 2
**Rating:** 4
**Confidence:** 4

**Summary:**

In this work, the authors introduce a framework for training visual language models (VLMs) that can produce effective sequence of actions on a mobile phone's GUI to complete user-provided task. The authors first argue that previous methods designed for this task are trained on offline data of specific environments which hinders their performance in an unseen environment. They propose a training framework that identifies various possible tasks on a mobile, designs a curriculum based on the complexity of those tasks, models the rewards to avoid late successes and early failures, and trains a modification of popular GRPO algorithm. Authors compare their framework with previous methods and show consistent improvement. They also present an ablation study on high-level choices used in the framework to provide evidence for importance of individual choices.

**Strengths:**

1) I like how the authors designed this entire framework: the automatic curriculum design ad reward design.
2) I can see the potential of someone with physical disabilities use voice-assistance to control their mobile phone as long as safety is ensured. The algorithm proposed in this paper can be highly useful to personalize an AI model for individual users.
3) The paper shows that the proposed framework achieves significant improvements over prior techniques in terms of success rates and sample complexity.

**Weaknesses:**

**Major (my reason for not providing higher score)**
1) No examples of the inference of the trained system. The example shown in the appendix in the section D does not mention the details of the user request or explain whether the trajectory generated is of a trained AI model. Also, the paper does not provide any details about the identified tasks through their method of synthetic task generation.
2) There is no comparison of proposed trajectory-aware MobGRPO against standard GRPO. If I understand correctly, the proposed algorithm uses length of the observations in the batch of data to normalize the total per-token loss. There are a few other algorithmic variations like Dr. GRPO [1] which can also be used for the comparison study.
3) Tables 2 and 3 report average success rates of different models attempting the GUI control task. How are these rates computed? How many number of attempts were allowed? Is there an analysis on success rate and attempts? Also, if it is possible, I would like to see the standard error across the test examples as it would provide me sufficient information to judge the statistical significance of these results.
4) The figure 2 shows that there is instability in learning with the curriculum proposed in this work. In the description, authors suggest this instability is not reflective of the model's true task proficiency. A more convincing argument would be to show how exactly the true performance of the model changes as the training progresses, i.e., having a validation curve would immensely clarify any doubts. It would be a good addition to have training curves for different variations of the algorithm.
5) Why do you call the metric "impossible task ratio"? Shouldn't the curriculum keep this metric growing steadily instead of sudden increase? If the tasks are impossible, why are they shown to the model in the first place? Is their forgetting of the earlier skills, if completely new task that does not use any previously learnt technique appears as part of the curriculum?


**Minor:**
1) Line 81: "..reward-driven learning difficult.." I think authors intend to say success criterion based training is difficult. The current phrasing makes it sound as if authors are proposing a training mechanism that does not depend on rewards.
2) Line 90 says "four mobile agent benchmarks", however the tables only depict 3. Am I misunderstanding something?
3) Line 144: The novelty of the parallel environment is contestable. Creating virtual environments in reinforcement learning is a standard workflow in expediting training. In addition, I do not think your current setting requires a parallel env in the first place because the main bottleneck is LLM inference, not the phone emulator.


**General comment:**
If the major weaknesses highlighted above along with the minor ones and ethical concerns are satisfactorily addressed, I am willing to increase my rating of this work.

**Questions:**

I have asked my questions in the weakness section above.

**Details Of Ethics Concerns:**

I want to highlight, in general, it is possible that without appropriate safety measures, AI model can take control of the personal device to harm human interests. I suggest authors add their perspective on why developing such AI agents is still necessary for the current times. I also suggest them to comment on the possibility of safety violations induced by RL training of the AI model.

---

### Official Review · Reviewer_Qt87 · 2025-10-31

**Soundness:** 2
**Presentation:** 3
**Contribution:** 2
**Rating:** 4
**Confidence:** 4

**Summary:**

This paper introduces MobileGUI-RL, a scalable framework for training vision-based mobile GUI agents using online RL rather than traditional offline approches. The framework features a synthetic task creation pipeline leveraging self-exploration and task filtering via a text-based world model, and adapts GRPO to GUI navigation with trajectory-aware advantage estimation and composite rewards. Experiments on AndroidWorld, Android-in-the-Wild (General tasks and webshop) with Qwen2.5-VL-7b/32b as training base model, with further ablation studies examining the contributions of curriculum learning and reward design.

**Strengths:**

-	The experiment results show improvements for both Qwen2.5-VL 7b/32b.
-	The paper shifts to online RL with mentioning scalable data collection environments.
-	The case study offers an intuitive illustration of the agent fixed the failure that the pre-trained LLMs made.

**Weaknesses:**

-	Several directly relevant recent works, such as DigiRL and DistRL are not compared, despite clear overlap in online RL settings and robust GUI agents. This absence undermines the claims of uniqueness and makes positioning statements in Section 2 unsupported.
-	Insufficient details on key components are missing. For example, (1) the scalable environment is not well described. How to ‘align compute-intensive environment simulation with CPU and model training with GPUs’? (2) The text-based world model is not precisely described. How is the textual state structured? How to deal with compounding errors? (3) the reward design details, such as decay parameter $\lambda$, clipping factors, penalty constant, are not empirically justified. (4) It is not clear how the tasks are created in curriculum.
-	The paper over-claims a few contributions including task self-exploration (initially mentioned in voyager [G.Wang 2023], and similar to task generations in OS-Genesis, GUI-Xplore, ScreenAgent), GRPO modification with trajectory-level and token-level information (already used in DAPO).

**Questions:**

See the weakness.

---

### Meta-Review · Area_Chair_uWSQ · 2026-01-05

**Summary:**

The paper introduces MobileGUI-RL, a framework for training vision-based mobile GUI agents using online Reinforcement Learning (RL). The method proposes a synthetic task creation pipeline involving self-exploration and task filtering via a text-based world model. It also adapts the GRPO algorithm to GUI navigation by incorporating trajectory-aware advantages and composite rewards. The authors evaluate the approach on AndroidWorld and Android-in-the-Wild benchmarks.

**Reviewer Concerns:**

Since there was no author response, the significant concerns raised by the reviewers remain unaddressed. The primary reasons for rejection include:

Missing Baselines and Comparisons: Reviewer Qt87 noted the absence of comparisons to directly relevant recent works such as DigiRL and DistRL, which undermines claims of uniqueness. Reviewer h578 highlighted the lack of comparison between the proposed "trajectory-aware MobGRPO" and standard GRPO or other variants (e.g., Dr. GRPO), making it difficult to attribute performance gains to the specific algorithmic modifications.

Insufficient Methodological Details: Multiple reviewers (Qt87, HP2q, QBaE, ojrd) pointed out critical missing details required for reproducibility. These include specific hyper-parameters for the reward design (decay, clipping), the structure of the text-based world model, the mechanism for scalable environment simulation, and the exact construction of the curriculum/task filtering.

Novelty Claims: Reviewers questioned the novelty of certain contributions, such as self-exploration (previously explored in Voyager) and parallel environment execution, which are standard in RL.

Presentation and Stability: Reviewers noted instability in learning curves and requested better analysis of the model's true performance progression.

**Reviewer Scores:**

Reviewer Qt87 (Score: 4): The reviewer's concerns regarding missing comparison with related work and insufficient details were not addressed. Their score would likely remain unchanged or decrease.

Reviewer h578 (Score: 4): The reviewer requested inference examples and baseline comparisons which were not provided. The score would remain unchanged.

Reviewer HP2q (Score: 2): The reviewer identified significant issues with reproducibility and experimental rigor. With no response, their recommendation for rejection would stand firm.

Reviewer QBaE (Score: 4): The reviewer required clarification on writing, definitions, and training details. The score would remain unchanged.

Reviewer ojrd (Score: 4): The reviewer had questions regarding data sources and sampling efficiency that remain unanswered. The score would remain unchanged.

---

### Decision · Program_Chairs · 2026-01-26

Reject